# Encoding Hierarchical Information in Neural Networks helps in Subpopulation Shift

## Abstract

Over the past decade, deep neural networks have proven to be adept in image classification tasks, often surpassing humans in terms of accuracy. However, standard neural networks often fail to understand the concept of hierarchical structures and dependencies among different classes for vision related tasks. Humans on the other hand, seem to learn categories conceptually, progressively growing from understanding high-level concepts down to granular levels of categories. One of the issues arising from the inability of neural networks to encode such dependencies within its learned structure is that of subpopulation shift – where models are queried with novel unseen classes taken from a shifted population of the training set categories. Since the neural network treats each class as independent from all others, it struggles to categorize shifting populations that are dependent at higher levels of the hierarchy. In this work, we study the aforementioned problems through the lens of a novel conditional supervised training framework. We tackle subpopulation shift by a structured learning procedure that incorporates hierarchical information conditionally through labels. Furthermore, we introduce a notion of graphical distance to model the catastrophic effect of mispredictions. We show that learning in this structured hierarchical manner results in networks that are more robust against subpopulation shifts, with an improvement of around 2% in terms of accuracy and around 8.5% in terms of graphical distance over standard models on subpopulation shift benchmarks.

## 1 Introduction

Deep learning has been tremendously successful at image classification tasks, often outperforming humans when the training and testing distributions are the same. In this work, we focus on tackling the issues that arise when the testing distribution is shifted at a subpopulation level from the training distribution, a problem called subpopulation shift introduced recently in BREEDS (Santurkar et al., 2021). Subpopulation shift is a specific kind of shift under the broader domain adaptation umbrella. In domain adaptation, the task of a classifier remains the same over the source and target domains, but there is a slight change in the distribution of the target domain (Goodfellow et al., 2016). In the general setting, the target domain is a slightly changed version of the source domain. For example, an object detector that has been trained to detect objects during day-time for a self-driving car application is used to perform the same task, but now on a shifted set of night-time images. The task remains the same i.e. to identify and detect objects, but the target domain (night-time) is a shifted version of the source domain (day-time), provided all other conditions (such as weather, region, etc.) remain constant.

However, in the setting of subpopulation shift, both the source and target domains remain constant. The shift here occurs at a more granular level, that of subpopulations. Consider the source distribution described above, that of a self-driving car. Let's say the categories for classification included small-vehicles and large-vehicles. Under small-vehicles, the source set included samples of golf-car and race-car, and under large-vehicles, the source samples were from firetrucks and double-decker-buses. In the target domain for testing, the classes remain unaltered; the classifier is still learning to categorize vehicles into small or large categories. However, the testing samples are now drawn from different subpopulations of each class which were not present during training, such as coupe and sedan for small-vehicles and dumpster-truck and school-bus for large-vehicles.

Given that this shift occurs naturally at the subpopulation level, which is one level of hierarchy below the classes to be classified, explicitly incorporating the notion of hierarchies among categories should help make the classifier more robust to this shift. Taking inspiration from humans, we want our neural networks to learn in a systematic manner, learning coarse categories first and then successively refining them down to more fine-grained categories Skinner (1958), also exploited in the field of curriculum learning as introduced in Bengio et al. (2009). This allows the network to encode inter-dependencies that a model trained on a flat set of classes cannot incorporate. Knowledge of these inter-dependencies helps under subpopulation shift, since the model has explicitly been made aware of a larger distribution at different levels covering many classes, and we hope that shifted unseen subpopulations will overlap with these larger covers at the higher levels.

Additionally, the current way of classification, in which each class is considered separate and independent of others, treats all mispredictions as having the same impact. This is contrary to intuition, since a Husky and a Beagle are not as equidistant to each other as they are to a Bull-frog. The impact of misclassifications becomes quite important in critical use cases. The cost of mispredicting an animate object for an inanimate object can be disastrous for a self-driving car. To address this, we introduce the notion of catastrophic coefficient, which is a quantitative measure of the impact of mispredictions that follows intuitively from our hierarchical graph. It is defined as the normalized shortest path distance between the true and the predicted classes, as per the graphical structure of the underlying hierarchy. We show that incorporating hierarchical information during training reduces the catastrophic coefficient of all considered datasets, both with and without subpopulation shift.

We explicitly incorporate the hierarchical information into learning by re-engineering the dataset to reflect the proposed hierarchical graph, a subset of which is sketched out in Figure 1. We modify the neural network architectures by assigning intermediate heads (one fully connected layer) corresponding to each level of hierarchy, with one-hot labels assigned to the classes at each level individually, as shown in Figure 2. We ensure that only samples correctly classified by a head are passed on for learning to the next heads (corresponding to descendants in the hierarchy graph) by a conditional learning mechanism. This allows the same neuron at a head to represent different classes conditioned on the previous head. We show results on a custom dataset we create out of ImageNet, and a subpopulation benchmark dataset introduced by BREEDS (Santurkar et al., 2021). We also show results on the BREEDS dataset by keeping the hierarchical structure, but changing the target subpopulations to cover a more diverse range. We show that given a hierarchy, our learning methodology can result in both better accuracy and lower misprediction impact under subpopulation shift.

To summarize, Our main contributions are as follows:

- We mitigate the effect of subpopulation shift by incorporating hierarchical information in two ways: 1) allowing independent inference at each level of hierarchy by encoding classes at that level in a one-hot fashion and 2) enabling collaboration between these levels by introducing a conditional training mechanism that trains each level only on samples that are correctly classified on all previous levels.

- We introduce the notion of misprediction impact, and quantify it as the shortest graphical distance between the true and predicted labels for inference.

- Evaluate the performance of deep models under subpopulation shift and show that our training algorithm outperforms classical training in both accuracy and misprediction impact.

- Re-engineer the ImageNet dataset to create a custom dataset that reflects inter-class hierarchies. Additionally, we show results on an existing benchmark dataset on subpopulation shift to show that our learning algorithm is compatible with any hierarchy.

## 2 RELATED WORK

**Subpopulation Shift** is a specific variant of domain adaptation where the models need to adapt to unseen data samples during testing but the samples arrive from the same distribution of the classes, changed only at the subpopulation levels. This setting differs from zero-shot learning since the classes remain the same. BREEDS (Santurkar et al., 2021) introduced the problem of subpopulation shift along with tailored benchmarks constructed from the ImageNet (Deng et al., 2009) dataset. WILDS (Koh et al. (2021)) provides a subpopulations shift benchmark but for toxicity classification

across demographic identities. Cai et al. (2021) tackle the problem through a label expansion algorithm similar to Li et al. (2020) but tackles subpopulation shift by using the FixMatch ( Sohn et al. (2020)) method. The algorithm uses concepts such as pseudo-labelling and consistency loss to learn visual concepts in a semi-supervised manner. Cai et al. (2021) applied the same and showed how consistency based loss is suitable for tackling the subpopulation shift problem. But these semi-supervised approaches require access to the target set, albeit unlabelled, and the algorithm makes use of these unlabelled target set to further improve upon a teacher classifier. We restrict ourselves to the supervised training framework where we have no access to the target samples. Moreover, we tackle the subpopulation shift problem by incorporating hierarchical information to the models.

**Hierarchical Modeling** is a tried and tested supervised learning strategy to learn superior concepts in vision datasets. Yan et al. (2015) introduce HD-CNN, which uses a base classifier to distinguish between coarser categories whereas for distinguishing between confusing classes, the task is pushed further downstream to the fine category classifiers. Deng et al. (2010) showed that the classification performance can be improved by leveraging semantic information as provided by the WordNet hierarchy. Deng et al. (2014) further introduced HEX graphs to capture semantic relations between two labels (parent and children). Although this work relabels leaf nodes to intermediate parent nodes, they train models only on the leaf node labels (single label). Our work on the other hand leverages the information at multiple levels of a hierarchy and employs a conditional training approach to link multiple labels of a single image as per the provided hierarchy. Works such as McClelland et al. (2016) and Saxe et al. (2013) tried to understand the importance of hierarchical learning from a theoretical perspective and demonstrated an implementation on a neural network based model. Jiang et al. (2019) proposes a Conditional class-aware Meta Learning framework that conditionally learns better representations through modeling inter-class dependencies. Blocks (Alsallakh et al., 2018) demonstrate how learning hierarchies is an implicit method of learning for convolutional neural networks. Song & Chai (2018) shows how multiple heads of a neural network can collaborate among each other in order reach a consensus on image classification tasks. Tree-CNN (Roy et al., 2020) tackles the incremental learning problem where the model expands as a tree to accommodate new classes. Contrary to these works, we utilize hierarchy as a way to mitigate the effect of subpopulation shift, both under accuracy and impact of mispredictions.

A recent work, Bertinetto et al. (2020) introduced a similar notion of catastrophic impact on mispredictions. This work leverages the graphical distance and optimizes this directly as a part of the loss function. On the other hand, we use the misprediction distance only as an evaluation metric to quantify the impact of our conditional training framework on the degree of catastrophic predictions across domain shifts during inference. We show that our models have lower catastrophic predictions than standard models under the subpopulation shift. Additional related work has been listed in A.3.

## 3 HIERARCHIES TO MITIGATE THE EFFECT OF SUBPOPULATION SHIFT

### 3.1 SUBPOPULATION SHIFT

As described in Section 1, subpopulation shift is a specific branch of the broader domain adaptation problem. In subpopulation shift we have training and testing distributions that differ at the level of subpopulations. We now introduce the required notation. Let's say we are focusing on an n-way classification problem, with each class denoted by $i$; $i = \{1, 2...n\}$. The data consisting of image-label pairs for the source and the target domain are denoted by $\{\mathbb{X}^s, \mathbb{Y}^s\}$ and $\{\mathbb{X}^t, \mathbb{Y}^t\}$, respectively.

Now we introduce notation for subpopulations. Each class $i$ draws from $s$ different subpopulations. The different subpopulations of class $i$ for training (source domain) are denoted by $S_i^s$ and for testing (target domain) by $S_i^t$. We reiterate that between source and target domains, the $n$ classes remain the same, since the classification task is unchanged. However the data drawn for each class at the subpopulation level shifts, with no overlap between the source and target subpopulations. This reflects that the subpopulations used for testing are never observed during training, i.e. $S_i^s \cup S_i^t = \varnothing$.

### 3.2 HIERARCHICAL VIEW TO TACKLE SUBPOPULATION SHIFT

In this work we tackle the subpopulation shift problem by explicitly incorporating hierarchical knowledge into learning via labels. Intuitively, if a neural network can grasp the concept of structural hierarchies, it will not overfit to the observed subpopulations. Instead, the network will have a notion of multiple coarse-to-fine level distributions that the subpopulation belongs to. The coarser

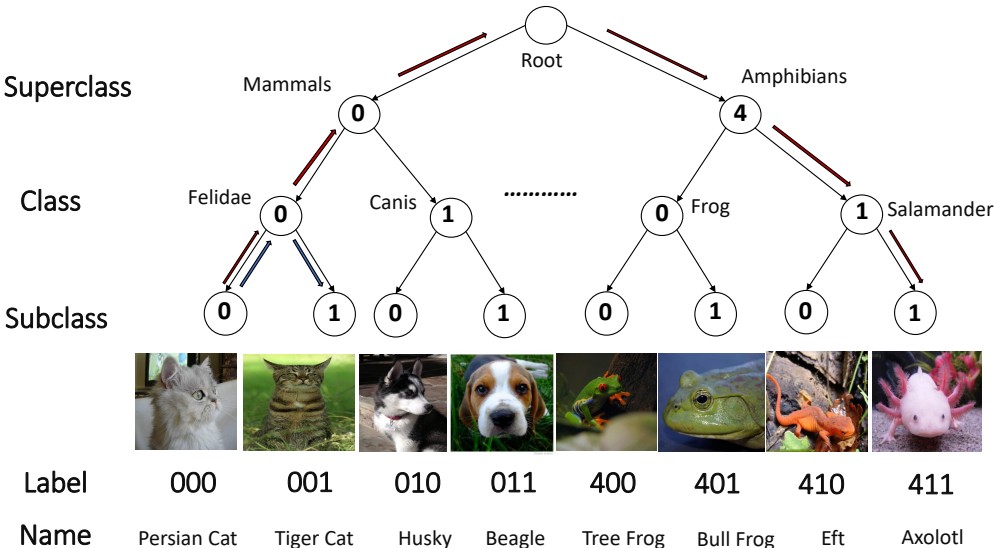

Figure 1: Hierarchical representation of a custom subset of ImageNet. The classes for the classification task are at the intermediate level, denoted by 'class'. The constituent subpopulations of each class are particular classes from the ImageNet dataset and are marked at the leaf level as 'subclass'. One-hot labels are provided at each level of the tree. The labels at the leaf nodes encode the overall hierarchical information. The colored arrows indicate the graphical distance from one leaf node to the other. This shows that mispredicting a Persian Cat as a Tiger Cat (two graph traversals) is far less catastrophic than predicting the same as an Axolotl (six graph traversals).

distributions would likely cover a much larger set of distributions, hopefully helping in generalization under shift. For instance, a network trained with the knowledge that both a fire-truck and a race-car fall under vehicles, and a human and a dog fall under living things, will not overfit to the particular subpopulation but have a notion of vehicles and living things. This will allow it to generalize to a newer large-vehicle such as school-bus and predict it as a vehicle rather than a living thing, since the network has learned a much broader distribution of vehicles one level of hierarchy above. Even if there is a misprediction, it is more likely to be at the lower levels of hierarchy, confusing things that are less catastrophic to mispredict.

### 3.3 VISION DATASETS AS HIERARCHICAL TREES

ImageNet (Deng et al., 2009) is a large-scale image database, which is collected on the basis of an underlying hierarchy called WordNet (Miller, 1992). It consists of twelve different subtrees created by querying synsets from the WordNet hierarchy. To motivate the problem of subpopulation shift, we create two custom datasets from ImageNet, which are subpopulation shifted versions of each other. The datasets have a balanced hierarchical structure of depth 3 as shown in Figure 1, starting from coarse concepts such as mammals and amphibians at a higher level, to fine grained specific subpopulations at the leaf nodes.

The hierarchical structure has 5 nodes at the highest level of superclasses, 10 nodes at the class level ($n = 10$), and each class draws from 3 subpopulations each ($s = 3$), leading to a total of 30 classes per dataset for source and target datasets each. Figure 1 shows two subtrees from the dataset, and two out of the three subclasses at the leaf nodes per class. During testing under shift, the 10 classes at the class level remain the same, but the 30 subpopulations that samples are drawn from are changed. Accuracy and catastrophic coefficients are reported for the 10 classes, similar to BREEDS.

The custom trees are simple and balanced, capturing the hierarchical structure found in the dataset. We use them to lay the foundations on which we implement our conditional training framework.

Later, we show how our method translates well to a complicated hierarchy such as the LIVING-17 (Santurkar et al., 2021) subpopulation shift benchmark. This illustrates that our algorithm is compatible with any hierarchy chosen according to the task of interest. Next, we show how to annotate the graphical hierarchical model to encode the concepts of hierarchy.

## 3.4 Hierarchy Encoding

Given a tree, we start from the root node and traverse downwards until the first level of hierarchy, which consists of superclasses such as mammals, fish, reptiles, etc. The custom dataset for instance has five superclasses and we label them as $0-4$. Next we traverse downwards till the level of classes. These are the actual tasks that the network has to classify. In this level we capture slightly finer concepts conditional to the previous level, such as given an amphibian, is it a frog or a salamander; or given a bird, is it aquatic or aviatory. Each superclass in our custom dataset has only 2 classes, getting labeled 0 or 1, making up the $n = 10$ classes for classification. Thus the categorical label of frog in our tree shown in Figure 1 is $40$, with the label $4$ encoding that it is an amphibian and the overall $40$ encoding that given it is an amphibian, it is a frog. Finally we reach the leaf nodes of the tree, where there are three subclasses per class, annotated from $0 - 2$. At the leaf nodes, we have specific classes from ImageNet, encoded in our level-wise concatenated format. Thus we have a one-hot encoded label for each level of our hierarchy, where each level captures the structural concept associated with the tree. The overall encoding conveys the information as a path arising from the superclasses and ending at the subclasses. More concretely, an encoding states that a Siberian Husky, annotated $010$ is a mammal (0), is a member of the Canis family (01) and it is a Siberian Husky (010). During training we split the encoded information into individual one-hot labels at each level so that we can train a head per level using categorical cross-entropy.

## 3.5 Catastrophic Distance

In this section, we introduce the concept of catastrophic coefficient as a measure of the impact of misprediction. It is the shortest graphical distance between the true label and the predicted label in our hierarchy, normalized by the number of samples. It implicitly quantifies whether there is a notion of semantic structure in the model's predictions. Subpopulation shift occurs at a lower level of a hierarchical tree where unseen subclasses are introduced during evaluation. So, if the hierarchically trained networks can grasp the concepts of superclasses and classes, the mispredictions during the shift will not be catastrophic. This is because they will tend to be correct at the higher levels, and hence "closer" to the ground truth node in terms of graph traversal.

Neural networks trained via standard supervised learning have no explicit knowledge of inter-class dependencies. Thus, for flat models, mispredicting a specific sub-breed of a dog as a sub-breed of a cat is as catastrophic as mispredicting the same as a specific species of a snake. Graphical distance between the true and predicted classes intuitively captures the catastrophic impact of a misprediction and accounts for the semantic correctness of the prediction. This serves as an additional metric to accuracy for evaluating the performance of models under subpopulation shifts. Additionally, it illustrates that the improvement in accuracy is truly due to incorporating better hierarchical information, rather than model architecture changes or the conditional training framework. It is pictorially illustrated by the colored arrows in Figure 1. A higher graphical distance between a misprediction and its ground truth signifies a more catastrophic impact. We average the graph traversal distances of all predictions ($= 0$ if sample classified correctly) over the entire dataset and call it the catastrophic coefficient, thus quantifying the impact of mispredictions for a network-dataset pair.

Let $g_k$ be the graph traversals needed for sample k in the shortest path between its true and predicted label. Let there be $N$ samples for evaluation. Then, the catastrophic coefficient is defined as $Cat = \frac{\sum_{k=1}^{N} g_k}{N}$. We note that we use this distance just to evaluate, and not during training. Instead we just take the final classifier level predictions and run it via our graph to check for distances in both models, irrespective of whether they have been shown the hierarchy or not.

## 3.6 Architecture

In this subsection, we describe the modifications on standard ResNet (He et al., 2016) architectures to make them suitable for our conditional training framework. Our network makes classification decisions at each level of the hierarchy. Hence, we introduce a separate head to predict the one-hot encoded vectors at each level. For e.g. if we want to represent the hierarchical concept of a dog

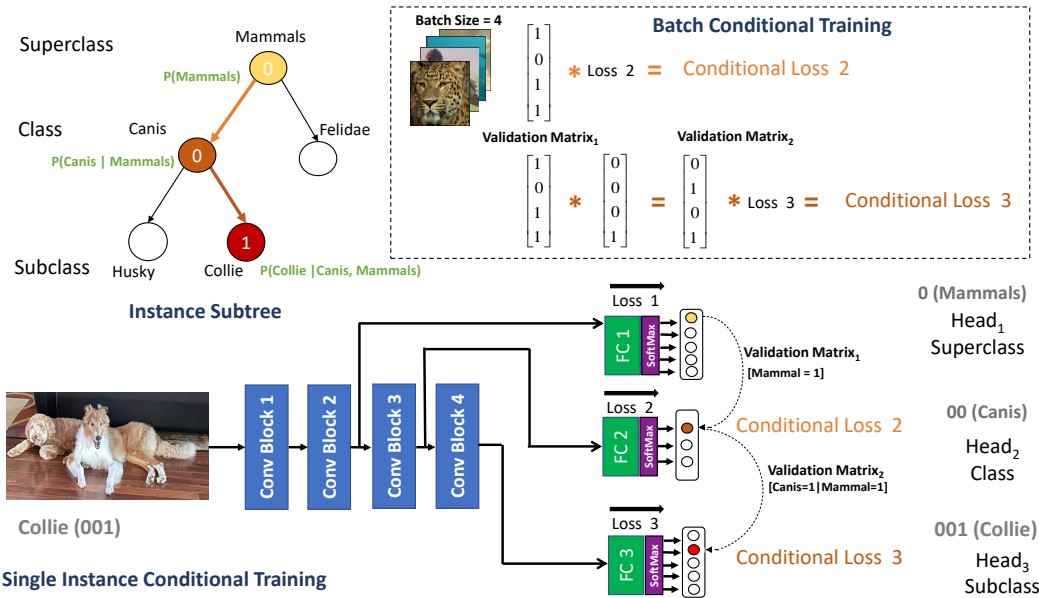

Figure 2: Figure shows our conditional training framework applied to a multi-headed neural network architecture, on the instance subtree shown on the top left. The bottom of the figure shows conditional training for a single instance of a class 'Canis', subclass 'Collie'. Each head predicts the hierarchical level-specific label from the instance subtree. The validation matrices dictate which samples get propagated for learning to the deeper heads. At the top right, the batch conditional training details are shown. Each element in a validation matrix specifies whether the sample is correctly predicted at a previous level. Here, the training is shown till the subclass level.

as shown in Figure 2, we want the $Head_1$ of our network to predict if it is a mammal (superclass), $Head_2$ to predict if it is a carnivore (class) and finally $Head_3$ to predict that it is a dog (subclass).

Convolutional Neural Networks learn coarse to fine features as we go deeper, capturing local concepts of images early on and global concepts as we go deeper (Zeiler & Fergus, 2014). This lines up well with our hierarchical structure, and hence we connect the different heads at different depths of the model. The concept is pictorially depicted in Figure 2. Since we use Residual Networks in our work, the individual convolutional blocks here are residual blocks. The locations of these heads are determined experimentally. We got best results with $Head_1$ attached after the third residual block, $Head_2$ and $Head_3$ after the fourth residual blocks. This maps the flow of coarser to finer categories in our hierarchy to the features needed for such classification. Our Hierarchy encoding enables us to reuse the same label at a particular level of the hierarchy to mean different categories, conditioned on the prediction at previous levels. This joint prediction mechanism requires collaboration between the multiple heads, and we ensure this via a conditional training approach which we describe next.

### 3.7 CONDITIONAL TRAINING DETAILS

Here we describe the conditional training framework, which is independent of subpopulation shift. Let us assume we have 3 levels in our hierarchy, with the levels enumerated by $l = 1, 2, 3$. Let the labels at each of these levels (treated as one-hot) be denoted by $y_l$. Let $F$ be the neural network that we pass a batch of images $X$ to. Here $X \in \mathbb{R}^{B \times d}$ where B is the batch-size and d is the dimension of the input data. Further, let $F_l$ be the neural network up to head $l$, corresponding to predicting at level $l$ of our hierarchical graph. Note that all $F_l$ have overlap since the neural network is shared, rather than an ensemble. This is illustrated in Figure 2. The loss at head $l$, $L_l$ is calculated as:

$$L_l = CrossEntropy(F_l(x), y_l) * V_{l-1}$$

where $V_l \in \mathbb{R}^B$ refers to the validity matrix generated at the the $l - th$ level, and $V_0$ is initialized to identity. This validity matrix is what enforces our conditional training framework. It is a batch-sized (batch-size for training) binary matrix, where each element is a 1 or 0, capturing whether the corresponding image was correctly classified by the $l - th$ head or not, respectively. Multiplying the previous head's validity matrix with the current head's loss before backpropagation ensures that

learning for $F_l$ only occurs on samples that are meaningful at that head. In other words, $V$ propagates only correctly classified samples, as per its name. This ensures that the prediction at the $l - th$ head is not just $p(y_l)$, but $p(y_l \mid F_{l-1}(x) = y_{l-1})$, or the prediction probability given that the previous head's prediction was correct. This is done because the same neuron will represent multiple classes while training each head individually, for instance the same neuron represents 'Felidae' and 'Frog' in Figure 1. However, this collaboration between heads via conditional training separates each neuron as conditioned on previous prediction and allows network to learn progressively refining predictions.

## 4    EXPERIMENTAL RESULTS

In Section 3, we described in detail our training framework starting from encoding hierarchies into supervised labels to conditionally train multi-headed networks. In this section, we empirically demonstrate how models trained with our approach perform better than models trained in a supervised learning setup, under the problem of subpopulation shift. We show results on three different datasets (two custom) and one BREEDS LIVING-17 subpopulation shift benchmark. We see that both in terms of accuracy and catastrophic coefficient, our hierarchical models are superior to standard ones in tackling the subpopulation problem. To cover a more diverse shift, we retain the hierarchy introduced in LIVING-17 but consider 2 more sets of subpopulations. We call these LIVING-17-B and LIVING-17-C and talk more about them in subsection 4.5.

### 4.1    SETUP

As mentioned, we consider subpopulation shift at the class level of a hierarchy. Consider an n-way classification problem, with each class denoted by $i$; $i = \{1, 2...n\}$. For our custom datasets, $n = 10$ and for the LIVING-17 dataset, $n = 17$. The total number of levels of hierarchy including the subpopulation levels, $l$, is 3 and 4 for our custom dataset and LIVING-17 respectively. Now, we create the shift by sampling subpopulations of each class $i$ from $s$ different subpopulations. For the two different datasets, $s = 3$ and $s = 2$ respectively. Thus, for the custom datasets we have a 10-way classification problem with a total of $n \times s = 30$ subpopulations, and 34 for LIVING-17. The $n$ classes are located at $l = 2$ in our custom tree and at $l = 3$ in LIVING-17, and the subpopulations at $l = 3$ and $l = 4$ respectively. We show the subpopulation details of our custom dataset in Appendix A.1 and we add the graphical representation as supplementary files.

More concretely, the subpopulations for class $i$ are distributed over $S_i^s$ (source) and $S_i^t$ (target) domain. Let's consider $i = \{\text{dogs}\}$, $S_{dogs}^s = [\text{Bloodhound, Pekinese}]$ and $S_{dogs}^t = [\text{Great-Pyreness, Papillon}]$. Thus the learning problem is that by training on just the source subpopulations $S_{dogs}^s$, the model will be able to say that the target subpopulations of $S_{dogs}^t$ belong to $i = \{\text{dogs}\}$.

### 4.2    TRAINING DETAILS AND METRICS

We use accuracy and catastrophic coefficient to measure performance, both in the presence and absence of subpopulations shift. Both catastrophic coefficients and accuracies are reported under 4 different settings, where applicable, differentiated by the subscript. The prefix of the subscript determines the domain the model was trained on and the suffix denotes the domain it is evaluated on. There are 4 combinations. '$s - s$' does not evaluate subpopulation shift, but shows the results of using our method as a general training methodology. It is trained on training source domain data, and evaluated on the validation set in the source domain. '$s - t$' evaluates results under subpopulation shift: training is performed on source domain, and testing on target domain. We interchange source and training domain and get the equivalent remaining two settings, '$t - t$' and '$t - s$'.

Throughout our experiments we focus mainly on two sets of training, which result in the Baseline and the Hierarchical Models. The Baseline models are trained in a flat manner, and evaluated as per (Santurkar et al., 2021). The classification task is on the $i$ classes, enumerated at the level of 'classes' mentioned in the hierarchy. The subpopulation shift occurs one level below, at the 'subclass' level. The subclass labels are never shown to the network, and hence one level of hierarchy is automatically incorporated. We remove this implicit notion of hierarchy in 'Subclass Level' models by training all the way down to subpopulations, so now the classification is between $n \times s$ subpopulations explicitly. The Hierarchical models, as the name suggests have been trained on the complete hierarchical information present in the tree, using our conditional training framework.We use ResNet-18 and

Table 1: Results on our Custom Dataset

|  | Source to Source (no shift) | | Source to Target (Shift) | |
|---|---|---|---|---|
|  | $\text{Acc}_{s-s}$ | $\text{Cat}_{s-s}$ | $\text{Acc}_{s-t}$ | $\text{Cat}_{s-t}$ |
| Hierarchical-18 | **87.2** | **0.88** | **52.0** | **2.51** |
| Baseline-18 | 84.79 | 0.89 | 49.53 | 2.70 |
| Subclass Level-18 | 85.84 | 0.97 | 46.91 | 2.84 |
|  | Target to Target (no shift) | | Target to Source (shift) | |
|  | $\text{Acc}_{t-t}$ | $\text{Cat}_{t-t}$ | $\text{Acc}_{t-s}$ | $\text{Cat}_{t-s}$ |
| Hierarchical-18 | **81.33** | **1.18** | **58.88** | **2.18** |
| Baseline-18 | 77.69 | 1.21 | 55.47 | 2.39 |
| Subclass Level-18 | 79.11 | 1.28 | 55.31 | 2.47 |

ResNet-34 as our network architecture backbones for modifications as mentioned in subsection 3.6. For enumerating results, we use 'Hierarchical-18' to denote a modified ResNet-18 model trained conditionally. Similarly 'Baseline-34' signifies a ResNet-34 architecture trained for flat classification on the $n$ categories found at the level of 'classes' in the hierarchy. We train the models on each dataset for 120 epochs with a batch size of 32, starting with a learning rate of $0.1$, and a 10 fold drop every 40 epochs thereafter. We do not use data augmentation on our custom datasets. All models have been trained on three random seeds each and the mean numbers are reported.

### 4.3 RESULTS ON CUSTOM DATASETS

In this section, we discuss the results on the Custom Datasets, shown in Table 1. For the custom datasets, we train our hierarchical models on the specific subpopulation classes so that they have knowledge of all the levels in the hierarchy. $Acc_{s-t}$ and $Cat_{s-t}$ denote the accuracy and catastrophic coefficient of the model during the $s-t$ shift. As shown in Table 1, the Hierarchical models perform better than the flat Baseline and Subclass level models in terms of accuracy and catastrophic coefficient both in the presence and absence of a shift. The performance gap is significant under shift. The models trained conditionally have $\sim 3\%$ improvement in terms of accuracy and a $7.03\%$ and $8.7\%$ improvement in terms of catastrophic coefficient over the flat baseline class level models under shift for the two custom sets ($s-t$ and $t-s$) respectively.

### 4.4 RESULTS ON BREEDS LIVING-17

In this section we discuss the results on the BREEDS LIVING-17 dataset, enumerated in Table 2. This subpopulation shift benchmark, introduced in (Santurkar et al., 2021), captures finer details of hierarchy that encode richer relationships between the entities. We show that our methodology is applicable to complex hierarchies and outperforms flat baseline models both on $Acc_{s-t}$ and $Cat_{s-t}$. Contrary to our custom datasets, in which we let the conditional models train down to the specific subpopulations, here we follow the BREEDS evaluation procedure and train all models only till the class level, never exposing models to subpopulation level labels. We train each architecture and model on five random seeds each and report the mean numbers. Contrary to BREEDS, we report numbers without bootstrapping, but follow all their other hyper-parameters. Hierarchical models achieve an improvement of around $2\%$ and $1.5\%$ in terms of accuracy and around $8.5\%$ and $7.36\%$ in terms of catastrophic coefficient for ResNet-18 and ResNet-34 respectively, as seen in Table 2.

**Additional Experiments**. We perform experiments on two additional sets of models, the 'Subclass Level' models, which are the flat models trained with subpopulation level labels, and the 'Separated' models. For the separated model, we remove the collaborative conditional training and train each head *separately*, keeping everything else the same. It achieves $Acc_{s-t}$ and $Cat_{s-t}$ values of $56\%$ and $1.72$ respectively compared to $59.02\%$ and $1.60$ for the hierarchical models. The performance drop justifies the use of validity matrices and conditional training for collaborative learning.

The 'Subclass Level' model has $Acc_{s-t}$ and $Cat_{s-t}$ values of $54.63\%$ and $1.83$ respectively. Additionally, as seen in Table 1, a Subclass model with no explicit information of a hierarchy has much

Table 2: Results on BREEDS LIVING-17 with and without shift

| Model | $Acc_{s-s}$ | $Cat_{s-s}$ | $Acc_{s-t}$ | $Cat_{s-t}$ |
|---|---|---|---|---|
| Subclass-18 | 91.59 | 0.36 | 53.71 | 1.86 |
| Baseline-18 | **92.3** | **0.33** | 57.02 | 1.75 |
| Hierarchical-18 | 92.1 | **0.33** | **59.04** | **1.60** |
| Baseline-34 | **92.56** | 0.33 | 59.82 | 1.63 |
| Hierarchical-34 | 92.32 | **0.32** | **61.32** | **1.51** |

Table 3: Results on BREEDS LIVING-17 as source, shifted to LIVING-17-B and -C as target

| Model | $Acc_{s-t}(B)$ | $Cat_{s-t}(B)$ | $Acc_{s-t}(C)$ | $Cat_{s-t}(C)$ |
|---|---|---|---|---|
| Baseline-18 | 53.54 | 1.97 | 53.04 | 1.93 |
| Hierarchical-18 | **54.35** | **1.86** | **54.14** | **1.81** |
| Baseline-34 | 55.86 | 1.84 | 55.52 | 1.812 |
| Hierarchical-34 | **55.92** | **1.79** | **55.62** | **1.77** |

worse shift performance than the baseline (single level hierarchy) and hierarchical (multi-level hierarchy) model, confirming that knowledge of hierarchy is helpful under subpopulation shift.

### 4.5 RESULTS ON SHIFTED LIVING-17

To understand the adverse effects of varying subpopulations we create two more shifted versions of the Target set of LIVING-17, LIVING-17-B and LIVING-17-C by varying the $S_i^t$ subclasses. $Acc_{s-t}(B)$ denote the model accuracy for the shift $s - t$ from set A to B. As can be seen from Table 3, Hierarchical-18 models have much better accuracy and catastrophic coefficients than the BREEDS baseline models for both shifted sets. This shows that our way of imparting hierarchical knowledge helps deep models to adapt to various adverse degrees of the subpopulation shift problem.

## 5 CONCLUSION

In this paper, we target the problem of subpopulation shift, which is a specific kind of shift under the broader umbrella of domain adaptation. Here, the subpopulations that make up the categories for the classification task change between training and testing. For instance, the testing distribution may contain new breeds of dogs not seen during training, but all samples will be labeled 'dog'. We note an implicit notion of hierarchy in the framing of the problem itself; in the knowledge of all constituent subpopulations sharing the common immediate ancestry. In line with this, we extend the notion of hierarchy and make it explicit to better tackle the issue of subpopulation shift. We consider the underlying hierarchical structure of vision datasets, in the form of both our own custom subset and a benchmark dataset for subpopulation shift. We incorporate this information explicitly into training via labeling each level with an individual one-hot label. We then encourage collaboration between multiple heads of a model via our conditional training framework, wherein each head is only trained on samples that were correctly classified at all levels before the present one. We further introduce a metric to capture the notion of semantic correctness of predictions. It uses the shortest graphical distance between the misprediction and the true label as per the hierarchy to quantify the catastrophic impact of mispredictions. We show that our hierarchy-aware conditional training setup outperforms flat baselines by around $\sim (2-3)\%$ in terms of accuracy and $\sim (7-8.5)\%$ in terms of catastrophic coefficient over standard models on custom and subpopulation shift benchmarks.

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

## A  APPENDIX

### A.1  CUSTOM DATASET

In this section, we provide a detailed description of the subpopulations $s$ of our custom datasets. We also provide the entire graph structure as separate supplementary files.

| Custom Dataset | | |
|---|---|---|
| Class | Source | Target |
| Felidae | Persian Cat, Tiger, Leopard | Egyptian Cat, Siamese Cat, Cheetah |
| Canis | White Wolf, Siberian Husky, beagle | Eskimo dog, Border Collie, Pomeranian |
| Small Fish | lionfish, puffer, goldfish | garfish, anemone fish, tench |
| Big Fish | Hammerhead, Tiger Shark, Great White Shark | Grey Whale, Killer Whale, Stingray |
| Salamander | common newt, European fire salamander, spotted salamander | axolotl, eft, spotted salamander |
| Frog | bullfrog, tree frog, tailed frog | bullfrog, tree frog, tailed frog |
| Snakes | Indian cobra, boa constrictor, green snake | hognose snake, rock python, green mamba |
| Non Snake | African crpcodile, mud turtle, Komodo Dragon | American alligator, leatherback turtle, Gila monster |
| Aviatory Bird | vulture, bald eagle, hummingbird | great gret owl, kite, redshank |
| Aquatic Bird | flamingo, black swan, king penguin | albatross, goose, pelican |

As can be seen there are 10 classes in the set and under each class, there are 3 subpopulations. Since these classes are obtained from the ImageNet (Deng et al., 2009) dataset, the total train dataset size is $1300 * 30 = 39000$ and the validation size $(s - s)$ or $(t - t)$ is 1500.

As seen from Table A.1, there is a partial overlap between the $S^s_{salamander}$ and $S^t_{salamander}$ and a complete overlap between $S^s_{frog}$ and $S^t_{frog}$. This is due to the fact that the ImageNet dataset does not have anymore Amphibian subpopulation classes. Thus, for fairness, while performing the $(t-s)$ or $(s-t)$ shift experiments we remove the amphibian subtree from the notion which consists of the classes, $i = \{$frogs, salamanders$\}$. Thus, the shift experiments it is a $8-$way classification problem rather than a $10-$way classification problem during $(s-s)$ or $(t-t)$.

## A.2 Hierarchical labels for BREEDS LIVING-17

We present the hierarchical labels for LIVING-17 dataset here and we have added the graphical structure as a supplementary file.

| LIVING-17 Labels | | | |
|---|---|---|---|
| Class | Source | Target | Labels |
| salamander | eft, axolotl | common newt, spotted salamander | 100 |
| turtle | box turtle, leatherback turtle | loggerhead, mud turtle | 210 |
| lizard | whiptail, alligator lizard | African chameleon, banded gecko | 220 |
| snake | night snake, garter snake | sea snake, boa constrictor | 200 |
| spider | tarantula, black and gold garden spider | garden spider, wolf spider | 300 |
| grouse | ptarmigan, prairie chicken | ruffed grouse, black grouse | 010 |
| parrot | macaw, lorikeet | African grey, sulphur-crested cockatoo | 000 |
| crab | Dungeness crab, fiddler crab | rock crab, king crab | 310 |
| dog | bloodhound, Pekinese | Great Pyrenees, papillon | 410 |
| wolf | coyote, red wolf | white wolf, timber wolf | 411 |
| fox | grey fox, Arctic fox | red fox, kit fox | 412 |
| domestic cat | tiger cat, Egyptian cat | Persian cat, Siamese cat | 413 |
| bear | sloth bear, American black bear | ice bear, brown bear | 414 |
| beetle | dung beetle, rhinoceros beetle | ground beetle, long-horned beetle | 320 |
| butterfly | sulphur butterfly, admiral | cabbage butterfly, ringlet | 321 |
| ape | gibbon, orangutan | gorilla, chimpanzee | 400 |
| monkey | marmoset, titi | spider monkey, howler monkey | 401 |

## A.3 Additional Related Work

**Domain Adaptation** and its variants are a well studied set of problems in deep learning. One direction of works (Ben-David et al. (2006), Saenko et al. (2010), Ganin & Lempitsky (2015), Courty et al. (2017), Gong et al. (2016), Donahue et al. (2014), Razavian et al. (2014)) is aimed at tackling the problem of adapting to target domains by learning on a selective set of samples from the target domain itself. Another line of work aims to match the source and target distributions in the feature space (Glorot et al. (2011), Ajakan et al. (2014), Long et al. (2015), Ganin et al. (2016)). The main motive behind these set of works is to tackle out-of-support domain adaptation tasks by sharing a common representation between the two. To adapt to newer environments, deep models are trained gradually to make them more suitable for transition to these newer environments (Gopalan et al. (2011), Gong et al. (2012), Glorot et al. (2011), Kumar et al. (2020), Chen et al. (2020)). Domain generalization enables the use of multiple different environments during training, but requires having a prior knowledge on the target distribution (Ghifary et al. (2015), Li et al. (2018), Arjovsky et al. (2019), Ye et al. (2021)). We on the other hand, focus on a more specific problem of distribution shift, wherein the shift occurs at a subpopulation level in the target domain.

**Hierarchical Modeling** Dhall et al. (2020) and Dhall (2020) show how an image classifier augmented with hierarchical information based on entailment cone embeddings outperforms flat classifiers on an Entomological Dataset. B-CNN (Zhu & Bain, 2017) learns multi-level concepts via a branch training strategy on low dimensional datasets. Condition CNN (Kolisnik et al., 2021) improves upon the training time of B-CNN and shows improved classification results on Kaggle

Fashion Product Images dataset. Pham et al. (2021) takes advantage of the relationship between diseases in chest X-rays to learn conditional probabilities through image classifiers. Taoufiq et al. (2020) adapts a similar approach to learn urban structural relationships. The following works that tackle the problem of learning hierarchical concepts on large scale image datasets ( Zheng et al. (2017) and Qu et al. (2017)). SS-HCNN (Chen et al., 2019) applies a semi-supervised approach to learn cluster level concepts at higher level of a hierarchy and categorical features at leaf node levels. However, these works do not focus on the problem of subpopulation shift.

## A.4 LEARNING MODELS

We introduce the different learning strategies we train the neural networks on. We show results on 4 sets of models, described below.

- Baseline Models - These are the primary models for evaluating subpopulation shift. The classification task is on the $i$ classes, enumerated at level of 'classes' mentioned in the hierarchy. The subpopulation shift occurs one level below, at the 'subclass' level. The subclass labels are never shown to the network, and hence one level of hierarchy is automatically incorporated. The results shown in (Santurkar et al., 2021) follow this methodology.

- Hierarchical Models - These are models resulting from training with our conditional framework described in this work. As the name suggests, the models have been trained on the complete hierarchical information present in the tree. Thus these models have the concepts of all levels of the hierarchy i.e superclasses, classes and subclasses. This is a collaborative way of knowledge representation where you do not need separate models each trained on separate levels of the hierarchy.

- Subclass Level Models - These models are trained on the individual subpopulation classes, for example on given $S_{dogs}^s = [Bloodhound, Pekinese]$. These models lack even the one level of hierarchy available to class level models. Strictly speaking, the shift in distribution between source and target is no longer subpopulation shift, since we are already exposing the subpopulation labels to the network, making them the classification classes. Here, the shift is a more general evaluation of what happens when the models are queried with domain shifted examples of each class.

- Separated Models - These models have the concepts of hierarchy as the hierarchical models but they are not trained under the conditional training framework. The individual heads mentioned in the subsection 3.6 are all trained separately with no conditional loss or validity matrix. In subsection 4.4, we show that such models perform poorly despite of having knowledge of the underlying hierarchy. Hence, these models corroborate the need for conditional training under our hierarchical setup, irrespective of whether subpopulation shift occurs or not.

## A.5 RE-WEIGHING SOFTMAX PROBABILITIES TO TARGET IMBALANCE OF LIVING-17

Although the BREEDS LIVING-17 dataset catches the rich hierarchical relationships among different entities in the tree, it suffers from the issue of dataset imbalance at various levels of the tree. As can be seen from the graph for LIVING-17 attached as supplementary data, there is only one salamander class under the superclass amphibian, whereas under the mammal superclass there are seven classes. This kind of imbalance is noticed at other parts of the tree as well. To mitigate this problem, we re-weigh the soft-max weights of each of our heads. Since a specific head is responsible for the prediction of a particular level of our tree, re-weighing each softmax probability at a particular head does not interfere with the re-weighing of the same at a different head. Through the re-weighing, we want to give more weightage to the nodes that are responsible to output classes that occur less in the dataset and less weightage to the nodes which predict the more frequently occurring classes. We do this by using the following formula, $W_i = \frac{\sum_{k=1}^{N} n_k}{c * n_k}$. Here the numerator sums to the total number of samples in the dataset, making the normalization factor for inverse proportionality. In the denominator, the term $n_k$ refers to the number of samples for each class. And $c$ refers to the total number of classes for a particular head. For example, $W_0 = \frac{44200}{5*11*2600}$ refers to the soft-max node responsible to predict the label $0$ at the third level of the BREEDS LIVING-17 hierarchical tree. The numerator represents the dataset size. In the denominator, $5$ represents the number of classes ($c$) at

the third level of the tree. The number $11 * 2600$ represents the number of samples with label 0. The label 0 appears 11 times at the third level as per our hierarchy encoding. 2600 is the number of samples per class such as salamander, grouse, etc each of which has label 0.

## A.6   LIVING-17-B AND C

It is very difficult to incorporate all subpopulation classes in a particular dataset. To understand the adverse effects of varying subpopulations we create two more shifted versions of the Target set of LIVING-17, LIVING-17-B and LIVING-17-C by varying the $S_i^t$ subclasses.We do this either by adding disjoint subclasses of the ImageNet (Deng et al., 2009) or by creating different combinations of the existing $S_i^t$ with new disjoint subclasses. We reuse some of the $S_i^t$ subclasses due to the unavailability of the same in the ImageNet database. All the subpopulations of $i = \{\text{wolf}\}$ from the ImageNet database have already been covered in the $S_{wolf}^s$ and $S_{wolf}^t$ set, so we just reuse the $S_{wolf}^t$ in the sets B and C.

We present the hierarchical labels for extended LIVING-17 dataset here.

| LIVING-17 Labels | | | |
|---|---|---|---|
| Class | Source | Target | Labels |
| salamander | European fire salamander, spotted salamander | European fire salamander, common newt | 100 |
| turtle | terrapin, loggerhead | terrapin, mud turtle | 210 |
| lizard | Komodo dragon, American chameleon | frilled lizard, common iguana | 220 |
| snake | green snake, Indian cobra | sidewinder, horned viper | 200 |
| spider | barn spider, black widow | barn spider, black widow | 300 |
| grouse | partridge, quail | partridge, quail | 010 |
| parrot | African grey, sulphur-crested cockatoo | African grey, sulphur-crested cockatoo | 000 |
| crab | hermit crab, king crab | hermit crab, king crab | 310 |
| dog | dalmatian, otterhound | Australian terrier, German shepherd | 410 |
| wolf | white wolf, timber wolf | white wolf, timber wolf | 411 |
| fox | red fox, kit fox | red fox, kit fox | 412 |
| domestic cat | tabby, cougar | cheetah, tiger | 413 |
| bear | koala, brown bear | ice bear, koala | 414 |
| beetle | tiger beetle, leaf beetle | tiger beetle, leaf beetle | 320 |
| butterfly | monarch, lycaenid | monarch, lycaenid | 321 |
| ape | siamang, baboon | siamang, baboon | 400 |
| monkey | proboscis monkey, squirrel monkey | colobus, macaque | 401 |

