# OpenReview forum: "Encoding Hierarchical Information in Neural Networks Helps in Subpopulation Shift"
_ICLR.cc/2022/Conference — ICLR 2022 Submitted_

### Official Review · Reviewer_1XvT · 2021-11-02

**Correctness:** 3
**Technical Novelty And Significance:** 2
**Empirical Novelty And Significance:** 3
**Recommendation:** 5
**Confidence:** 4

**Main Review:**

I find the ideas presented are clear and straightforward to understand, and the evaluation results convincing.
The results are unsurprising given how the custom setup better addresses the goal of generalizing to subpopulation shift.
It is still informative to have a formal quantification of such generalization potential, although, such contribution renders the work as a limited experiment to address a custom problem or evaluate a specific question rather than a major contribution.

To this extent, my main concern is about the novelty of the presented work. In fact, several aspects claimed to be novel have been addressed several time in the literature over the past decade.
In particular, the idea of quantifying "misprediction impact" / "catastrophic coefficient" is very similar to the “hierarchical cost” by Deng et al. (ECCV 2010), the followup Hierarchy and Exclusion graphs (ECCV 2014).
Also, the proposed architecture is similar in spirit to HD-CNN (Yan et al. CVPR’15).
I expected the authors to reference the above pieces of work and to compare their solution and results against them.
The authors did reference other related pieces of work, again, without comparing against them (dismissing the need by mentioning that "Contrary to these works, we utilize hierarchy as a way to mitigate the effect of subpopulation shift"). Nevertheless, these pieces share several aspects of the proposed solution even if the exact tasks they address are slight different.

With that, the current contribution is rather incremental. I would have deemed it otherwise had the authors provided theoretical insights about the problem of hierarchy, (e.g. as in the work by McClelland, Saxe and others).

References:

Deng, J., Berg, A. C., Li, K., & Fei-Fei, L. “What does classifying more than 10,000 image categories tell us?." European conference on computer vision. Springer, Berlin, Heidelberg, 2010.

Deng, J., Ding, N., Jia, Y., Frome, A., Murphy, K., Bengio, S., Li, Y., Neven, H. and Adam, H. “Large-scale object classification using label relation graphs” In European conference on computer vision (pp. 48-64). Springer, 2014.

McClelland, J. L., Z. Sadeghi, and A. M. Saxe. "A Critique of Pure Hierarchy: Uncovering Cross-Cutting Structure in a Natural Dataset." Neurocomputational Models of Cognitive Development and Processing: Proceedings of the 14th Neural Computation and Psychology Workshop. 2017.

Saxe, Andrew M., James L. McClellans, and Surya Ganguli. "Learning hierarchical categories in deep neural networks." Proceedings of the Annual Meeting of the Cognitive Science Society. Vol. 35. No. 35. 2013.

Yan, Z., Zhang, H., Piramuthu, R., Jagadeesh, V., DeCoste, D., Di, W., & Yu, Y. “HD-CNN: hierarchical deep convolutional neural networks for large scale visual recognition”. In Proceedings of the IEEE international conference on computer vision (pp. 2740-2748), 2015.

Language issues:
- at a the subpopulation
- One hot labels => One-hot labels
- if a neural networks
- are lesser catastrophic => less
- Each heads predicts
- Our networks makes
- a batchsized binary matrix => unclear
- are superior than standard => superior to
- at level of => at the level of
- the the notion
- subclasses from the ImageNet => of ImageNet
- We now introduce some notation. => avoid vague statements involving “some”.

Formatting issues:
- Then the catastrophic coefficient, is defined => Then, …
- individually , => no space
- End all sentences with a full stop (e.g. the bullet points on page 2).
- for Toxicity classification => why capitalized?
- (Koh et al. (2021)) => avoid nested parentheses.

**Summary Of The Paper:**

The authors address the problem of generalization under subpopulation shift where the labels remain identical in the new domain, however, their instances belong to different subcategories than in the source domain. The authors propose infusing knowledge both about these subcategories and about super-categories of the labels as additional supervision signals. For this purpose, they define a conditional training framework along with an adapted architecture and a hierarchical loss. The evaluation results demonstrate how their framework significantly improves robustness to subpopulation shift.

**Summary Of The Review:**

The contribution is incremental, and many of the claimed novelty overlaps with the existing work (e.g. the references listed in the main review).

---

> ### Author Response · Authors · 2021-11-23
> **Addressing the comments of Reviewer 3. Please let us know if you have any further queries.**
>
> - Regarding related literature and comparisons
>
> We thank the reviewer for pointing us in useful directions with the references. We have added them to the updated manuscript, with the changes highlighted. Due  to time constraints during the rebuttal phase, we were unable to recreate the work in the context of subpopulation shift and compare their results with ours. However, we are grateful for finding related literature that validates our metrics and architecture.
>
> - Regarding formatting and language issue
>
> We thank the reviewer for pointing out the formatting and language issues and apologize for the oversight. We have fixed the errors in the updated manuscript and have highlighted the changes.

---

> > ### Comment · Reviewer_1XvT · 2021-11-24
> > **Novelty issue unresolved**
> >
> > I appreciate the clarifications by the authors and their effort to revise the manuscript.
> >
> > W.r.t. my comments, the revision is limited to citing additional references and to brief comments on how the presented work differs from them. I still find second item in the contribution list (the notion of misprediction impact) incremental compared with the work by Deng et al.
> > I understand that the metrics introduced by the authors are slightly different and are used in a slightly different setting. I also do think the experimental results can potentially inform future work in the area of encoding hierarchical information in neural networks.
> > Nevertheless, as it stand, the work IMO falls a bit short of being a solid contribution to ICLR'22.
> > Besides more comprehensive and comparative experimental results, the above-mentioned area would benefit form deeper theoretical understanding of the role of hierarchical information and accordingly a systematic way to leverage this information to improve the robustness to subpopulation shift (or other desired characteristics).

---

### Official Review · Reviewer_TcMm · 2021-11-02

**Correctness:** 3
**Technical Novelty And Significance:** 2
**Empirical Novelty And Significance:** 2
**Recommendation:** 3
**Confidence:** 4

**Main Review:**

**Strengths**
 - The problem is of great interest to the vision community.
 - Paper is well motivated, and method is technical sound.

**Weaknesses**
 - The proposal hierarchical classification method is not completely novel. For instance, [1] [2] proposed similar approaches / architectures for the hierarchical classification problem.
* I have a particular concern of the per-level parameter sharing design. This design requires the class hierarchy be perfectly balanced (i.e. each parent class at the same level has the same number of children classes). The authors may want to comment on how the proposed approach apply to these non-perfect but practically-common situations.
 - Experimental results are insufficient.  It would be more convincing if the authors include comparison with other hierarchical classification approaches in the literature. Also, limiting the method to subpopulation shift problems seem unnecessary.


[1] HD-CNN: Hierarchical Deep Convolutional Neural Network for Large Scale Visual Recognition

[2] Making Better Mistakes: Leveraging Class Hierarchies with Deep Networks

**Summary Of The Paper:**

This paper presents a new hierarchical classification method. Instead of using a flat classifier layer for all classes, the proposed approach trains a neural network with a hierarchy of classifier layers (attached to different blocks of the DNN), where each classifier layer corresponds to a specific level in the class hierarchy, and models the conditional classification probability given classes at parent level. The authors use the proposed method to tackle the subpopulation shift problem. Empirical results on a custom ImageNet dataset and the BREEDS dataset show improvements over baseline approaches.

**Summary Of The Review:**

Overall, contributions of the paper seem small. The proposal approach is not extremely novel and is limiting in practical settings (non-balanced class hierarchy). Experimental results lack comparison with literature hence not convincing.

---

> ### Author Response · Authors · 2021-11-23
> **Addressing the comments of Reviewer 2. Please let us know if you have any further queries.**
>
> - Regarding related literature and comparisons
>
> We thank the reviewer for pointing us in useful directions with the references. We have added them to the updated manuscript, with the changes highlighted. Due  to time constraints during the rebuttal phase, we were unable to recreate the work in the context of subpopulation shift and compare their results with ours.
>
> - Regarding imbalanced trees in the hierarchy
>
> We thank the reviewer for pointing out the oversight in clarifying the way we handle imbalanced datasets. Our custom dataset indeed has balanced trees , which is indeed a simpler problem than the more practical case where different classes contain a different number of examples. We face this issue when we extend our method to the hierarchies outlined by BREEDS. We tackle this by reweighing the softmax probability at each head by the proportion of samples it contains. Classes having fewer subclasses get their losses upweighted in comparison to the classes having more subclasses. We have added this to the appendix A.5 in the updated manuscript and have highlighted the changes.
>
> - Regarding limiting the method to subpopulation shift
>
> Our target in this manuscript was the problem of subpopulation shift since we find it highly relevant to the vision community. We agree that the method can be generalized to other areas of concern such as incremental learning,  unsupervised domain adaptation and zero shot learning with a few modifications.  We reserve extensions for future work and focus on subpopulation shift in this manuscript since we believe that this is a problem we are likely to encounter both often and with increasing impact.

---

> > ### Comment · Reviewer_TcMm · 2021-11-29
> > **Comment to authors' rebuttal**
> >
> > I would like to thank the authors for providing additional references and explanation for the class-imbalanced case. However, my concerns regarding the per-level parameter sharing and experimental results were largely unresolved. Therefore, I'm keeping my original rating.

---

### Official Review · Reviewer_GFoK · 2021-11-08

**Correctness:** 3
**Technical Novelty And Significance:** 2
**Empirical Novelty And Significance:** 2
**Recommendation:** 5
**Confidence:** 4

**Main Review:**

Strengths:
- This paper is well written. The architecture and evaluation methods are clearly explained and the research problem is well motivated.
- The proposed architecture is simple and computationally inexpensive which means it could be easy applied to any object recognition problem.
- On the datasets tested the authors demonstrated a performance benefit from using their training method.

Weaknesses:
- The proposed mulitheaded architecture is only evaluated on datasets containing images and a label hierarchy of living things. It may be the case that this limited range of object types have properties such as similar degrees of intraclass variability and the presence of distinct textures compared to other object types. Further experiments are required to prove that there are performance benefits to using this architecture.
- I would like to see more discussion and experiments evaluating the impact of the particular class hierarchy chosen for your custom dataset on your new models performance. As a baseline, what happens if subclasses are assigned to class groups randomly, what if the hierarchy is determined by visual features instead of semantic meaning (ex. feathers, scales, fur)? More experiments in this direction would provide a more convincing argument that your performance improvements are not only a result of the particular set of classes and hierarchy structure that you chose to use.
- In table 1, why is the source to source and target to target accuracy lower for Baseline-18 than it is for Subclass Level-18? Shouldn't we expect the opposite trend?
- Is there past work you could cite that supports the statement: "Humans on the other hand, seem to learn categories conceptually, progressively growing from understanding high-level concepts down to granular levels of categories." Couldn't humans also learn fine-grained categories first and then build them up into more abstract groupings?

**Summary Of The Paper:**

1. Introduces a new multi headed architecture for explicitly incorporating class hierarchy into training.
2. Created a new metric to measure the impact of a miss-prediction according to class hierarchy tree distance
3. Created a new label hierarchy for a subset of ImageNet
3. Evaluated the performance of their new architecture under subpopulation shift

**Summary Of The Review:**

Overall, this paper demonstrates some promise that there could be an advantage to explicitly specifying a label hierarchy during training. More experiments are necessary to demonstrate this method really works more generally on a wider variety of object types and label hierarchies.

---

> ### Author Response · Authors · 2021-11-23
> **Addressing the comments of Reviewer 1. Please let us know if you have any further queries.**
>
>
> - Regarding breadth of chosen hierarchical structures
>
> We thank the reviewer for raising the factor of similarity in our evaluated hierarchies. We accept that this is a common theme between the hierarchies we chose to evaluate, primarily stemming from the application perspective of evaluating the impact of misclassifying living objects, which (at least generally) appears to have a higher impact than misclassifying non-living objects. We accept that this is application dependent, and it would definitely be worthwhile to explore different hierarchies. Due to the lack of time during the rebuttal, were not able to train networks for different hierarchies, and hence we draw attention to the results spanning different complexities of hierarchies in our work rather than the different types.
>
>
>
>
> - Regarding evaluation with random hierarchies
>
> To understand the impact of hierarchy, we did run experiments with hierarchies that are semantically meaningless created by randomly allotting leaf classes to our fixed hierarchy at the top 2 levels. We noted that the accuracy of our method dropped considerably, but we were not surprised at this result since our method relies on the explicit externally provided hierarchy being mapped into the features learned by the DNN via the conditional training method. Randomly allotting classes breaks down feature share-ability and undermines the ability of the conditional training framework to learn common coarse to fine grained level features. We use BREEDS to demonstrate that our network can scale to different complexity of hierarchies, but we anticipate the need for hierarchies to be meaningful and to have commonalities that can be exploited in order to be learnt.
>
>
>
>
> - Regarding Baseline vs Subclass level accuracy
>
> We concur with the reviewers that the general expectation would be for the Baseline model to outperform the Subclass level model. We add results for the Subclass-18 model on BREEDS dataset as well in Table 2 in the updated manuscript with the changes highlighted. We only see this counter-intuitive trend for source to source (accs-s) and target to target (acct-t) experiments for the custom dataset, wherein there is no shift. The single level of hierarchical information encoded by the subclass model does not mean anything without shift. On the other hand, we do not see the counterintuitive trend wherever any shift is involved, as anticipated.
>
>
>
>
>
> - Regarding the justification of biological/human comparison of learning coarser to finer grained categories
>
> We take inspiration from the field of curriculum learning (Benjio et al, 2009) in machine learning, which is in turn inspired from animal training techniques known as shaping (Skinner, 1958). Here, tasks are introduced in order of complexity in order to guide training. The idea lends itself intuitively to hierarchical structures in which fine grained information is learned after the network inherits knowledge of coarser categories. We thank the reviewer for pointing out the need for corroboration here and have added the references to the paper.
>
> Skinner, B. F. (1958). Reinforcement today. American Psychologist, 13, 94–99.
>
> Benjio, Y et al. (2009). Curriculum learning. Proceedings of ICML, 2009

---

### Author Response · Authors · 2021-11-23
**Thanking all the reviewers.**

The authors wish to thank all the reviewers for their time and informative comments; it helped us improve our paper and exposed us to more, relevant literature. We have updated our manuscript according to the feedback received and the changes are addressed in red.

---

### Decision · Program_Chairs · 2022-01-20

**Decision:**

Reject

**Comment:**

The paper studies subpopulation shift in object recognition when classes obey a hierarchy. It proposes an architecture, a relevant metric and a dataset (subset of imagenet).  The problem of classification in hierarchical label spaces is important and of great interest, and the effect on domain shift is interesting. Naturally, this problem was studied quite intensively over the years.

Reviewers were concerned that the current proposal was not placed well enough in context of previous literature, both in terms of the method and in terms of experimental results.  Also, the paper would be strengthen if it provides more theoretical analysis about how the hierarchy helps with the domain shift. The authors addressed some of these issues in the rebuttal, adding references and highlighting the differences from previous methods, but the paper would need more time to make the proper experimental comparisons with previous work and subsequent analysis. As a result, the paper is still not ready for acceptance to ICLR in its current form.